# It Takes Two to Tango: The Interplay between Prostate Cancer and Its Microenvironment from an Epigenetic Perspective

**DOI:** 10.3390/cancers16020294

**Published:** 2024-01-10

**Authors:** Anniek Zaalberg, Elisabeth Pottendorfer, Wilbert Zwart, Andries M. Bergman

**Affiliations:** 1Division of Oncogenomics, The Netherlands Cancer Institute, Plesmanlaan 121, 1066 CX Amsterdam, The Netherlands; a.zaalberg@nki.nl (A.Z.); lisa.pottendorfer@gmail.com (E.P.); 2Laboratory of Chemical Biology and Institute for Complex Molecular Systems, Department of Biomedical Engineering, Eindhoven University of Technology, 5600 MB Eindhoven, The Netherlands; 3Oncode Institute; 4Division of Medical Oncology, The Netherlands Cancer Institute, Plesmanlaan 121, 1066 CX Amsterdam, The Netherlands

**Keywords:** prostate cancer, epigenetics, tumor microenvironment, androgen receptor targeted therapy, immunotherapy

## Abstract

**Simple Summary:**

Prostate cancer is a prevalent cancer in men. Metastatic disease is initially responsive to androgen receptor signaling inhibition, but eventually resistance develops despite continuation of therapy. This resistance involves multiple adaptations in the cell but also to the tumor microenvironment, including epigenetic alterations. Epigenetic alterations change gene expression without DNA sequence modifications and play a key role as regulators of cell functions both in the tumor and the tumor microenvironment. Moreover, these epigenetic alterations highlight potential therapeutic targets. Targeting these epigenetic modifications could improve androgen receptor-targeted therapy and enhance anti-tumor immunity. In this review we discuss the role of epigenetics in prostate cancer, strategies to target them, and their impact on the tumor microenvironment, with the goal of identifying novel therapeutic avenues for advanced prostate cancer.

**Abstract:**

Prostate cancer is the second most common cancer in men worldwide and is associated with high morbidity and mortality. Consequently, there is an urgent unmet need for novel treatment avenues. In addition to somatic genetic alterations, deviations in the epigenetic landscape of cancer cells and their tumor microenvironment (TME) are critical drivers of prostate cancer initiation and progression. Unlike genomic mutations, epigenetic modifications are potentially reversible. Therefore, the inhibition of aberrant epigenetic modifications represents an attractive and exciting novel treatment strategy for castration-resistant prostate cancer patients. Moreover, drugs targeting the epigenome also exhibit synergistic interactions with conventional therapeutics by directly enhancing their anti-tumorigenic properties by “priming” the tumor and tumor microenvironment to increase drug sensitivity. This review summarizes the major epigenetic alterations in prostate cancer and its TME, and their involvement in prostate tumorigenesis, and discusses the impact of epigenome-targeted therapies.

## 1. Introduction

Prostate cancer (PCa) is the second most diagnosed form of cancer among men worldwide [1]. Most patients present with organ-confined, localized disease, and diagnosis is often preceded by an increase in serum prostate-specific antigen (PSA) levels. Depending on the associated risk, patients may be monitored without intervention through active surveillance (mostly low-grade tumors) or can be treated with radiotherapy to the prostate or prostatectomy (removal of the prostate) with curative intent [2,3]. However, approximately 30% of these patients eventually develop biochemical recurrence—an increase in serum PSA levels—which is an indication of metastatic disease [2,4]. The treatment of these patients requires a systemic approach, with androgen receptor (AR)-targeted therapies as the most important pillar. Androgen deprivation therapy (ADT) represents the mainstay treatment for the patient population with metastatic disease who cannot be cured. ADT aims to suppress the testicular biosynthesis of androgens in order to inactivate AR signaling—the key driver of PCa initiation and progression [5]. AR represents a critical regulator in PCa cells and has been the focus of therapeutic development in recent decades. AR is a hormone-driven transcription factor essential for normal prostate development but also tumorigenesis [6,7]. Mechanistically, AR interacts with its natural ligand testosterone, dimerizes and translocates to the nucleus [7]. In the nucleus, AR binds the DNA at androgen response elements (AREs), driving the expression of androgen-responsive genes to promote cell survival and cell growth. Other transcription factors, such as FOXA1 and HOXB13, serve as pioneer factors for AR and render chromatin accessible to facilitate AR-driven transcriptional programs [8,9]. AR is expressed in virtually all primary prostate tumors, and remains expressed in most metastases [6,10].

Although practically all patients initially experience tumor regression following AR-targeted therapy, the response is only transient, ultimately leading to metastatic castration-resistant prostate cancer (mCRPC), which remains uncurable. The most prominent resistance mechanism in mCRPC involves restoration of the AR signaling axis and, consequently, continued dependence on androgen-signaling pathways, through somatic amplifications of the AR gene itself [11] and/or an upstream AR enhancer [12,13], or the expression of constitutively active AR splice variants (AR-SV), of which AR-V7 is the best studied [14,15]. In addition, in vitro studies have revealed other mechanisms by which PCa cells can acquire hormone therapy resistance, such as intratumoral androgen production or AR compensation through glucocorticoid receptor action [16]. Next to genetic alterations, epigenetic rewiring [17,18] and transdifferentiation are also mechanisms of resistance, including neuroendocrine (NE) prostate cancer, which occurs as an adaptive response under the pressure of prolonged AR-targeted therapy [19,20]. Therapy-resistant mCRPC highlights the need for new approaches to treat advanced disease. An alternative treatment approach currently being explored is based on the fact that cancer is a disease not only driven by genetic mutations, but also dictated by epigenetic alterations [21]. Recent studies have shown that the epigenomic landscape in mCRPC is remarkably reprogrammed compared to indolent treatment-naïve PCa [22,23]. These epigenetic modifications reprogram the cascade of various transcription factors and affect downstream gene expression, thereby driving PCa plasticity and neuroendocrine differentiation. Distinct from genetic mutations, epigenetic influences refer to modifying gene expression without permanent changes in the genomic sequence. Since epigenetic modifications are reversible and more rapidly altered compared to genetic changes, they are preferentially adopted by cancer cells, for example, in response to treatment or during disease progression [17,24,25,26]. Exploring the role of the epigenome in the development of treatment resistance may reveal epigenetic processes that are susceptible to novel therapeutic interventions [21] and lead to the discovery of unique diagnostic or predictive biomarkers [22,27].

The prostate tumor microenvironment (TME) plays a critical role in the development and progression of PCa, and altering the TME could potentially provide novel ways to treat patients and improve survival [28,29,30,31]. Furthermore, epigenetic alterations occur in tumor cells, but also in cells within the TME. Therefore, targeting epigenetic modifications may boost anti-tumor immunity and could enhance current therapies [4,32]. In this review, we will touch upon four main questions: (1) What are the roles of various epigenetic modifiers in PCa initiation, progression and aggressiveness? (2) How can epigenetic alterations be targeted in PCa? (3) What is the role of epigenetic aberrations in shaping the immunosuppressive properties of the prostate tumor microenvironment? (4) Are epigenome-targeting agents able to reverse tumor-associated immune evasion and therefore sensitize PCa to immunotherapy? We will collectively highlight which specific epigenetic alterations in tumor cells and the TME are emerging as potential targets for the treatment of advanced PCa and discuss promising avenues to explore for future therapeutic development.

## 2. What Are the Roles of Various Epigenetic Modifiers in PCa Initiation, Progression and Aggressiveness?

In embryonic development, the dynamics of epigenetic-mediated gene regulation plays a key role, and this type of gene regulation is frequently reactivated in cancer initiation and progression [33,34,35]. Epigenetic modifiers can be differentially expressed in tumor tissues compared to their normal counterparts, resulting in altered epigenetic profiles contributing to the initiation and progression of cancer [36]. An overview of important epigenetic modifiers affecting PCa is depicted in Figure 1 and discussed in further detail below.

### Histone Modifiers in Prostate Cancer Development, Progression and Treatment

Histone modifications and their mediators play a critical role in DNA transcription and replication, chromosome packaging and DNA repair [37,38], and deviations in their regulators promote prostate tumorigenesis [27]. For example, histone deacetylase (HDAC) overexpression can lead to gene silencing [39] and is associated with PCa progression and aggressiveness [40], with HDAC2 serving as an independent prognostic marker [41,42]. In addition, histone lysine demethylases (KDMs) are a group of histone-modifying enzymes that can remove both activating and repressive histone marks. KDMs are grouped into seven classes, each targeting a specific type of methylation. Histone lysine demethylase 1A and 4C (KDM1A and KDM4C) are associated with PCa progression and androgen-dependent proliferation [22]. KDM1A (LSD1) is overexpressed in primary tumor lesions and correlates with an increased risk of recurrence [21,43]. Mechanistically, KDM1A demethylates H3K9me1 and me2, often associated with heterochromatin, and interacts with AR to allow AR-mediated gene transcription [44]. KDM4C, which localizes to KDM1A, also demethylates H3K9me1-3 to enhance AR function, linking both KDM1A and KDM4C to androgen-dependent proliferation [21,22]. In addition, another histone lysine demethylase, KDM5D, has been shown to act as a tumor suppressor by physically interacting with AR in the nucleus and regulating its transcriptional activity by demethylating H3K4me3 active transcription marks, to suppress AR signaling [22,45].

Another histone-modifying enzyme that has been implicated in promoting tumorigenesis is Enhancer of Zeste 2 (EZH2), the catalytic subunit of polycomb repressive complex 2 (PRC2) and a key epigenetic repressor of gene transcription, which catalyzes the trimethylation of H3K27 (H3K27me3) [21,27,46]. In contrast, EZH2 also serves as a coactivator for several transcription factors, including AR [22]. EZH2 overexpression and amplification increase as tumors progress to metastatic disease and can lead to the aberrant silencing of tumor suppressor genes [27,47].

Another prominent example of the influence of histone-modifying enzymes on PCa development is the frequent dysregulation of bromodomain-containing (BRD) proteins [22,48]. BRD-containing proteins are chromatin readers with a variety of different catalytic and scaffolding functions within the cell. For example, BRD4 is a critical regulator of AR expression and is frequently upregulated in mCRPC. In addition, AR co-regulators p300 and CBP are paralogous proteins that are frequently increased in mCRPC and serve as transcriptional regulators. They contain bromo- and histone acetyltransferase (HAT) domains and are capable of acetylating histone H3 at lysine residue 27 (H3K27ac) [22,49], a mark for active enhancers and gene transcription [50,51,52]. In addition, p300 and CBP can acetylate AR at lysine 632 and lysine 633, enhancing the AR’s activation status [53]. The increased expression of BRD4, p300 and CBP results in the progression of AR-dependent PCa, and they are all associated with PCa tumorigenesis and poor prognosis [22,41], rendering these proteins interesting therapeutic targets.

Lastly, Nicotinamide N-methyltransferase (NNMT), a metabolic enzyme that catalyzes the methylation of nicotinamide using the universal methyl donor S-adenosyl-L-methionine (SAM), connects metabolism with epigenetic remodeling [54]. NNMT can influence several epigenetic enzymes, such as histone deacetylase sirtuins and NNMT expression, and is significantly positively correlated with both stromal and immune components, resulting in a tumor-promoting microenvironment [55,56,57]. Additionally, NNMT is frequently overexpressed and associated with a poor prognosis in various cancers, including prostate cancer [58]. NNMT impairs the methylation potential of cancer cells by consuming methyl units from SAM, thereby creating the stable metabolic product 1-methylnicotinamide, resulting in an altered epigenetic state that includes hypomethylated histones, contributing to tumorigenesis and the aggressiveness of tumor cells [54].

## 3. How Can Epigenetic Alterations Be Targeted in PCa?

Epigenetic alterations can be targeted using specific drugs that modulate the activity of enzymes involved in epigenetic regulation, such as HDACs. These drugs can restore epigenetic patterns by inhibiting aberrant histone modifications, leading to, amongst others, the reactivation of tumor suppressor genes and the suppression of oncogenes. Below, we describe several classes of drugs targeting specific epigenetic modifications.

### 3.1. Histone Deacetylase Inhibitors

As described above, HDAC inactivation has multiple effects on PCa tumors, including the induction of apoptosis and suppression of angiogenesis [21,59]. HDAC inhibitors induce cellular differentiation [60] as well as cell-cycle arrest [61] and the inhibition of AR-responsive gene transcription [62]. Furthermore, HDAC inhibitors can also inhibit the activity of proteins involved in cell proliferation and survival, such as E2F1 and p53, resulting in the stabilization of these proteins [63,64,65]. To date, four HDAC inhibitors (vorinostat, romidepsin, belinostat and panobinostat) have already been FDA-approved for the treatment of lymphoma and melanoma [22,66]. Notably, while vorinostat, belinostat and panobinostat are pan-HDAC inhibitors that target different HDAC members, romidepsin is an agent designed to selectively inhibit HDAC1 and HDAC2 [66]. Interestingly, drug–drug synergy was observed in cell lines and mCRPC PDX tumors in vitro between enzalutamide and vorinostat, providing a therapeutic proof-of-concept [67]. In addition, the HDAC3-selective inhibitor RGFP966 blocks the transcriptional activity of both full-length AR as well as its constitutively active splice variant, AR-v7 [57]. However, with respect to PCa, most HDAC inhibitors are currently still being evaluated in preclinical studies. Exceptions are romidepsin and vorinostat, which entered phase II clinical trials [27] that enrolled mCRPC patients before (romidepsin) or after (vorinostat) chemotherapy treatment (Table 1). However, the response was limited to a few patients with a partial response or short-lived stable disease. 

Unfortunately, all patients treated with vorinostat experienced toxicity, and nearly half of the patients treated with romidepsin discontinued therapy due to toxicity [68,69]. However, because in vitro studies suggest synergism between combined HDAC and AR inhibition, combination therapy has been evaluated in early clinical trials. Panobinostat was evaluated in a phase I/II trial together with the AR inhibitor bicalutamide in patients with recurrent PCa [66]. HDAC1-targeted therapy reduced AR-mediated resistance to bicalutamide in CRPC cell lines, with longer progression-free survival for patients treated with a combination of bicalutamide and panobinostat versus bicalutamide alone (Table 2) [22,70]. However, further studies on the combination of anti-HDAC agents with AR inhibitors are needed to draw further conclusions on the benefit of the combination.

### 3.2. EZH2 Inhibitors

EZH2 inhibition has been shown to induce interferon-stimulated genes (ISGs) by derepressing double-stranded RNA (dsRNA). This phenomenon is referred to as viral mimicry and involves the re-expression of dormant endogenous retroviral sequences (ERVs), transposable elements that are often repressed to prevent aberrant autoimmunity [76,77,78]. Since the presence of dsRNA in the cytoplasm is associated with viral infections, the transcription of dsRNA leads to the activation of antiviral signaling pathways and results in anti-tumor immunity and responsiveness to checkpoint inhibition therapy [79,80,81]. In murine and human PCa organoids, it has been shown that treatment with EZH2 inhibitors significantly induced intracellular levels of dsRNA [82]. In addition, human prostatectomy samples with low H3K27me3 and increased dsRNA levels had higher PD-L1 expression compared to patient samples with high H3K27me3 and low dsRNA expression [82]. Targeting EZH2 as a therapeutic strategy for primary PCa or CRPC is considered particularly interesting as EZH2 not only serves as coactivator of AR, but also binds to the AR promoter region to amplify its expression [83]. Consequently, most studies evaluating the use of EZH2 inhibitors in PCa have focused on combination therapies involving anti-EZH2 drugs and AR inhibitors. For example, a phase 1b/2 clinical trial is currently evaluating the effect of the EZH2 inhibitor CPI-1205 in combination with the AR inhibitor enzalutamide, the androgen biosynthesis inhibitor abiraterone or the glucocorticoid receptor agonist prednisone in mCRPC patients (Table 2) [20,66,71]. The same AR-targeted therapies are also being evaluated in a phase I clinical trial for their synergistic effects with tazometostat, another EZH2 inhibitor, in patients with mCRPC (Table 2) [22,66,72]. Since EZH2 inhibitors trigger a viral mimicry response, this provides a strong rationale for combining these drugs with immunotherapies.

### 3.3. BET, p300/CBP, LSD1 and NNMT Inhibitors

BET inhibitors target the bromodomain and extra-terminal (BET) family, a subset of BRD-containing proteins [48]. In PCa cell lines and xenografts, it has been observed that AR signaling is disrupted by BET inhibitors, with BRD4 no longer being recruited to chromatin [20,84]. Due to the effect of BET inhibitors on AR signaling and the resulting anti-tumorigenic effects, they have been investigated in particular in combination with AR inhibitors. One example is the BET inhibitor ZEN-3694, which was evaluated in a phase II clinical trial in CRPC patients in combination with enzalutamide (Table 2) [20,22]. This combination exhibited promising efficacy and tolerability in patients with mCRPC. These findings suggest the need for additional prospective studies, particularly in a subset of mCRPC cases characterized by low AR transcriptional activity [73]. In addition, p300 and CBP are frequently overexpressed in PCa, and their inhibition has shown anti-tumor activity in preclinical studies. A phase I/II clinical trial using the P300/CBP inhibitor CCS1477 has shown that AR and MYC signaling is downregulated (Table 2) [74,85]. In contrast, the evaluation of the efficacy of KDM1A (LSD1) inhibitors as a PCa treatment has not progressed beyond preclinical studies. KDM1A expression is often increased in PCa, and preclinical studies in cell lines have demonstrated that inhibiting KDM1A leads to anti-tumor activity [20,22]. For example, the inhibition of KDM1A was observed to prevent pioneer factor FOXA1 from binding chromatin, thereby decreasing AR activity and suppressing tumor growth. These observations make KDM1A a potentially interesting therapeutic target, although further studies are needed to confirm these findings [22].

In addition, small-molecule inhibitors of NNMT have been developed in recent years and considered as therapeutics for metabolic diseases, such as diabetes, obesity and fatty liver disease [86,87,88]. Furthermore, several NNMT inhibitors show promising results in targeting cancer. For example, a study using a mouse model of ovarian cancer showed decreased tumor burden and tumor cell proliferation in mice treated with an NNMT inhibitor [55]. Another study showed that the use of an NNMT inhibitor reduced tumor cell viability and induced cytotoxicity in 2D/3D clear cell renal carcinoma-derived tumor models [89].

In conclusion, epigenetic patterns correlate with clinical and pathological predictors of PCa phenotype and outcome, and ongoing clinical trials evaluating novel epigenetic therapies hold promise as potential treatment strategies for patients with advanced PCa.

## 4. What Is the Role of Epigenetic Aberrations in Shaping the Immunosuppressive Properties of the Prostate Tumor Microenvironment?

Epigenetic alterations play an important role in shaping the immunosuppressive properties of the prostate TME [90]. These aberrations can lead to alterations in gene expression patterns, resulting in the downregulation of immune-related genes and the upregulation of immunosuppressive factors [91]. In addition, epigenetic modifications can influence the recruitment and function of immune cells within the TME, ultimately contributing to immune evasion and tumor progression [92].

### 4.1. Epigenetics in the Prostate Tumor Microenvironment

PCa is derived from epithelial cells that originate from the luminal cell population of the prostate [93]. In healthy conditions, tissue organization depends on consistent crosstalk between epithelial cells and the surrounding stroma [3,58]. Prostate tumor development is related to several genetic and epigenetic aberrations that result in uncontrolled cell growth [94]. These epigenetic alterations affect not only tumor cells, but also cells in the surrounding stroma, resulting in genotypic and phenotypic alterations of stromal components [93]. In addition, critical features of this abnormal microenvironment, such as oxidative stress, acidosis and oxygen deprivation, affect the vasculature, the epigenetic landscape of cells in the TME and the extracellular matrix and lead to the recruitment of immune cells [94,95]. As a result, a pro-tumorigenic TME is formed that includes an extracellular/stromal matrix, host cells (fibroblasts, immune cells, pericytes, etc.) and soluble factors (cytokines/chemokines) [4,96]. Importantly, interactions between the TME and tumor cells play a key role in tumor progression, metastasis and aggressiveness, as well as the development of resistance mechanisms [95]. Therefore, in recent years, the TME has become an increasingly important area of focus in the search for novel therapeutic approaches to the treatment of PCa.

### 4.2. Current Therapies Targeting the Prostate Tumor Microenvironment

Most therapies targeting the prostate TME have focused on the adaptive immune system and the respective roles of immune cells in tumorigenesis. Cancer vaccines, stimulating the host immune system by enhancing its recognition of tumor-associated antigens (TAAs), are currently being investigated for their efficacy as a PCa treatment [4]. This strategy seems promising since PCa cells express different prostate-specific antigens, such as prostatic acid phosphatase (PAP), prostate-specific membrane antigens (PSMA) and prostate-specific antigens (PSA) [97]. A limited number of mechanistically different vaccines have been explored. One example is cell-based vaccines that use antigen-presenting cells to activate T-cells, triggering an immune response [4,98]. While most of the vaccines investigated failed to demonstrate the expected beneficial results, one cell-based vaccine, Sipuleucel-T targeting PAP, received FDA approval for the treatment of mCRPC in 2010 [97]. However, the use of Sipuleucel-T is considered controversial since Sipuleucel-T results in a limited increase in overall survival and no observed effects on PSA levels, symptoms or tumor burden [97,98]. In addition, immune checkpoint inhibitors have dramatically changed the treatment landscape of multiple cancer types but have shown limited success as a PCa treatment. Similarly, the recently developed chimeric antigen receptor T-cell (CAR-T) therapy, which relies on the use of autologous T-cells engineered to recognize tumor antigens, has shown limited success in clinical trials treating PCa patients thus far [4,97].

### 4.3. Challenges of Current Therapies

Overall, immunotherapies targeting the prostate TME have faced many challenges. PCa is widely described as an immunologically “cold” tumor, which generally exhibits an immunosuppressive state with limited infiltration of cytotoxic T-cells and low expression of tumor neoantigens as a result of the low mutational burden of PCa cells [4,97]. The low accessibility of cytotoxic CD8+ T-cells to the TME leads to the formation of an immune-evading microenvironment and results in innate resistance to checkpoint inhibitors [99]. In addition, the loss of major histocompatibility complex (MHC) classes I and II, often observed in PCa, results in the reduced infiltration of cytotoxic T-cells into the tumor, and it has been shown to be one mechanism of immune evasion in mCRPC cell lines and clinical specimens [100,101,102]. Furthermore, the presence of tumor-associated macrophages (TAMs) is commonly associated with anti-inflammatory responses [103]. In addition to the immune cell-associated properties that pose challenges for immunotherapies, other PCa-specific features increase the difficulty of targeting the prostate TME as a therapeutic strategy. For example, PCa has a low mutational burden, and thereby, decreased neoantigen expression compared to other tumor types [104]. This results in a potential lack of T-cell co-stimulation and activation in the prostate TME, which prevents the generation of a powerful immune response following antigen presentation, a key step in immunotherapy effectiveness [105]. In addition, chronic inflammation, which is often observed in PCa, leads to a persistent exchange of inflammatory cytokines, resulting in tumor cell proliferation and the further formation of an immunosuppressive microenvironment, recruiting TAMs and myeloid-derived suppressor cells (MDSCs) [106,107]. Finally, cancer-associated fibroblasts (CAFs) support immunosuppression through the secretion of soluble factors and matrix proteins [95]. CAFs maintain an immunosuppressive microenvironment by recruiting regulatory T-cells and TAMs and promote the development of chronic inflammation, as well as the induction of T-cell exhaustion, through the secretion of cytokines such as TGF-β [95,108].

In conclusion, PCa is characterized by an immunosuppressive microenvironment consisting of TAMs, MDSCs and CAFs, with limited cytotoxic T-cell infiltration due to low levels of neoantigens resulting from a low mutational burden and the loss of MHC classes I and II. Therefore, agents that reverse this immunosuppressive microenvironment or enhance anti-tumor properties may have therapeutic potential for PCa patients.

## 5. Are Epigenome-Targeting Agents Able to Reverse Tumor-Associated Immune Evasion and Therefore Sensitize PCa to Immunotherapy?

Interestingly, epigenetics is increasingly being explored to target the immune microenvironment. Here, we describe epigenetic therapies currently being explored in PCa, and how these therapies can sensitize PCa to immunotherapies.

### 5.1. Epigenetic Therapies Targeting the Prostate TME to Increase Tumor Immunogenicity

Combination therapies of epigenome-targeting drugs with AR inhibitors or chemotherapeutics have been proposed and are under investigation. Similarly, to overcome the challenges of targeting the prostate TME, many therapeutic strategies are now exploring the combination of immunotherapies with drugs that may enhance the susceptibility of PCa to immunotherapeutics. Recent studies have shown that epigenetic aberrations in tumor and immune cells are also drivers of pro-oncogenic immune dysfunction and the development of resistance mechanisms to immunotherapies [109]. The reversible nature of epigenetic modifications and their correlation with the level of tumor immunogenicity has led to the investigation of epigenetically targeted drugs to potentially reprogram immune evasion and to increase tumor susceptibility to current immunotherapies [110].

### 5.2. Enhancing Tumor Immunogenicity Using HDAC Inhibitors

Recent studies have shown that HDAC inhibitors target epigenetic aberrations in both cancer and cancer-associated cells, and additionally, confer immune-enhancing properties in immunologically “cold” tumors [110]. Therefore, HDAC inhibitors could potentially help to overcome resistance and increase the susceptibility of immunosuppressive tumors to immunotherapies. Antigen presentation by MHC class I molecules on the cell surface is essential for the activation of cytotoxic T-cells. To evade the immune response, tumors use epigenetic modulators, such as HDACs, to decrease and alter the expression of MHC class I on their cell surface, thereby preventing tumor recognition by T-cells [77,111]. By using HDAC inhibitors, the expression of antigen-presenting MHC class I molecules has been observed to be significantly increased in studies using various tumor cells, including PCa cell lines [77,112]. This increase in MHC class I expression could enhance antigen presentation, leading to increased cytotoxic T-cell recognition and activation. Furthermore, other MHC class I processing and presentation genes, which are also frequently decreased in cancer cells, such as TAP1/2 and LMP2, were shown to be increased after treatment with HDAC inhibitors [77]. It is important to note that all the observed effects of HDAC inhibitors on the antigen presentation machinery have only been studied in tumor cell lines, and further in vivo research is needed to confirm these findings. In addition, epigenetic modulators are used to induce the activation of antiviral signaling pathways. This treatment strategy is based on viral mimicry and leads to the promotion of type I and type III interferon signaling [79]. The activation of this signaling pathway leads to the secretion of pro-inflammatory signaling molecules and the upregulation of antigen processing/presentation on cancer cell surfaces. Therefore, the induction of “viral mimicry” would enhance innate and adaptive immune responses and increase tumor susceptibility to immunotherapies [113]. However, further in vivo studies and analyses of clinical trials are needed to draw more comprehensive conclusions. Another important factor mediated by epigenetic modifications that affects the susceptibility of the tumor and its TME to immune responses is soluble signaling molecules, such as chemokines. The tumor and TME often suppress pro-immunogenic chemokines using epigenetic modulators to prevent immune responses such as T-cell infiltration [77,114]. HDAC inhibitors increased T-cell chemokine expression and infiltration in lung cancer models, inhibited tumor progression and sensitized cancer cells to PD-1 inhibitors [115]. Again, further studies in other cancer models are needed to confirm these findings. Finally, HDAC inhibitors also affect MDSCs, which exhibit critical immunosuppressive properties, such as preventing T-cell proliferation and activation [4]. MDSCs are, amongst others, mediated by various epigenetic modifications that affect gene expression and chromatin structure by regulating transcription factor binding [116]. The induction and differentiation of MDSCs involves histone modifications, underscoring the impact of histone acetylation on MDSC cell populations. Treatment with HDAC inhibitors significantly decreases MDSC levels in the TME of prostate adenocarcinoma mouse models [117]. Furthermore, in mammary tumor-bearing mice, an increase in T-cells was observed in addition to a reduction in MDSCs numbers in the TME [118]. This decrease in MDSC levels could further reduce tumor-associated immunosuppression and sensitize tumors to checkpoint inhibitors.

Epigenetic modifications also play a role in regulating the innate immune response, particularly NK cells, which are cytotoxic cells [119] whose activity is often downregulated in the TME [4]. HDAC inhibitors affect the NK cell-mediated killing of tumor cells in two different ways. In a co-culture experiment, the treatment of colon cancer cells with HDAC inhibitors resulted in the direct upregulation of the NK activating receptor NKG2D on the surface of NK cells [77,120]. Mechanistically, HDAC inhibitors induced histone acetylation at gene promoter sites of the NKG2D receptor, resulting in increased NK cell binding activity. Other HDAC inhibitors have been shown to increase the expression of stress-inducing ligands, such as MICA, MICB and ULB-3, on the surface of tumor cells, thereby increasing the susceptibility of these tumor cells to NK cell-mediated cytolysis [78].

In addition, due to their role in the adaptive immune response, T-cells are the focus of most immunotherapeutic strategies. One of the most prominent T-cell populations affected by epigenetic modifications are Treg cells, whose immunosuppressive properties play a critical role in maintaining the immune-evasive TME [121]. However, targeting Treg cells with HDAC inhibitors is complicated because they have heterogeneous targets involved in Treg epigenetic mediation, resulting in a variable response [77]. Therefore, additional studies are needed to achieve a better understanding of the effect of HDACs on Treg cells. Besides Treg cell activation and differentiation, epigenetic modifications also play an important role in the T-cell differentiation processes leading to T-cell exhaustion [78]. The state of T-cell exhaustion is attributed to the overstimulation of T-cells by antigens, resulting in non-functional T-cells due to the loss of sensitivity to signaling molecules [122]. T-cell depletion promotes the immune-evading properties of the TME and is often associated with the development of immunotherapeutic resistance. Due to the involvement of epigenetic modulators in the process of chronic overstimulation, drugs targeting these aberrant epigenetic modifications may be able to reverse T-cell exhaustion and revitalize them. Consequently, the restoration of T-cell activity could therefore sensitize tumors to immunotherapies.

### 5.3. Enhancing Immunogenicity with EZH2 Inhibitors

The increased expression of EZH2 occurs in both tumor and TME-associated cells, such as T-cells [123], NK cells [124], regulatory T-cells [125] and macrophages [126]. The pro-tumorigenic effect of EZH2 on tumorigenesis is not limited to tumor suppressor gene silencing, but also includes oncogenic immune evasion [109]. Similar to HDACs, EZH2 is involved in MHC regulation. Cancer cells often downregulate these antigen-presenting complexes and their respective antigen processing pathways to evade immune recognition [127]. The EZH2-mediated epigenetic silencing of H3K27me3 plays a critical role in the regulation of the antigen-presenting machinery by suppressing both the activation and basal levels of MHC classes I and II (Figure 2) [128,129]. In contrast to the heterogeneous effects of HDACs on Treg cell populations, studies investigating Treg-mediated tumor immunosuppression observed a significant increase in EZH2 expression, specifically in tumor-infiltrating Treg cells [128]. This upregulation of EZH2 activity in Treg cells is associated with the maintenance of Treg cells and their inhibitory properties, thereby promoting pro-tumorigenic immune evasion [109]. Furthermore, EZH2 upregulation is also associated with the prevention of T-cell infiltration into the TME by suppressing pro-immunogenic chemokines or cytokines, such as CXCL9 and CXCL10 [114]. Importantly, the effects of aberrant epigenetic modulators, including EZH2 overexpression, are not limited to the adaptive immune response, but also affect the antigen-independent innate immune system, such as NK cells. During functional immune responses, NK cells recognize tumor cells and use their cytolytic activities to induce cell death [127]. The upregulation of EZH2 in both cancer cells and NK cells negatively affects this immune response in two different ways. On one hand, the upregulation of EZH2 in tumor cells allows the cells to escape the cytolytic activities of NK cells [109]. On the other hand, the aberrant expression of EZH2 in NK cells directly inhibits their differentiation and activation [130]. Due to these diverse effects of EZH2 overexpression in tumor cells and TME-associated cells, the inhibition of EZH2 using epigenome-targeting agents may lead to a decrease in immunosuppression, thereby sensitizing tumors to immunotherapies. In addition, anti-EZH2 agents may restore antigen presentation processes mediated by the re-expression of MHC classes I and II. Consequently, these effects would lead to T-cell activation as well as an increase in T-cell infiltration into the TME, thereby distributing pro-tumorigenic immune-evasive properties (Figure 2). Consequently, tumors treated with EZH2 inhibitors would be more susceptible to immunotherapies. Furthermore, EZH2 inhibitors could prevent the negative effects of EZH2 overexpression on NK cells, thereby enhancing NK cell activation and differentiation and restoring their cytolytic activity to enhance anti-tumor immunity [109,124]. In addition, EZH2 activity is significantly lower in normal cells than in tumor cells and is particularly important during embryonic development [111]. As cells mature and differentiate, EZH2 loses its functional importance, making EZH2 a cancer cell-specific therapeutic target. An ongoing phase I/II clinical trial is evaluating the tolerability and efficacy of the EZH2 inhibitor CPI-1205, which inhibits EZH2 catalytic activity by binding to EZH2, in combination with the anti-CTL-4 checkpoint inhibitor ipilimumab in patients with advanced solid tumors (Table 2) [109,131]. Ongoing clinical trials and extensive preclinical studies will provide further insight into the ability of EZH2 inhibitors to sensitize immunologically “cold” PCa to immunotherapies. However, further research is needed to determine the efficacy, potential side effects and exact role of EZH2 in PCa.

### 5.4. Boosting Anti-Tumor Immunity Using NNMT Inhibitors

NNMT is expressed in tumor cells as well as cells in the TME [56]. The NNMT-dependent metabolite of nicotinamide, 1-MA, has tumor-promoting and immune-suppressing effects in ovarian cancer [132]. 1-MA that is secreted by NNMT-expressing fibroblasts and tumor cells is taken up by T-cells in the TME. In response, T-cells secrete increased tumor necrosis factor-α (TNFα) and decreased interferon-γ (IFNγ), ultimately leading to decreased cytotoxicity and an increase in tumor growth. In line with this, in a clear cell renal carcinoma model, NNMT expression has been correlated with the amount of Treg cells in the tumor tissue [89]. In addition, 1-MNA, alone and in combination with TGFβ, increases the amount of PD1-expressing CD4 T-cells and, to a lesser extent, CD8 T-cells [89]. Although further experimental evaluation is needed, this shows that 1-MNA acts as an immune-suppressive metabolite in clear cell renal carcinoma and ovarian cancer [89,133]. The first data in mouse models did not report adverse effects of NNMT inhibition or NNMT knockdown [75,133,134]. Furthermore, NNMT-derived peptides that are specifically represented by HLA molecules on tumors could allow for the immunologic targeting of NNMT-expressing primary tumors and metastases, in addition to molecular targeting [135].

## 6. Conclusions and Future Perspectives

This review summarizes the impact of epigenetic modifications on prostate tumorigenesis and the promising efficacy of targeting the epigenome. Many preclinical studies have identified epigenetic modifications as attractive therapeutic targets due to their potential reversibility and heterogeneous effects on pro-tumorigenic mechanisms. Targeting epigenetic alterations may simultaneously affect cellular mechanisms and signaling pathways in tumor and tumor-associated cells, as well as innate and adaptive immune responses. In PCa, HDAC inhibitors are the most widely studied epigenetic drugs, and significant progress has been made in their development over the past decades. However, clinical trials evaluating the efficacy of HDAC inhibitors often fail to match the promising results shown in earlier performed preclinical studies due to their heterogeneous effects. HDACs are a superfamily consisting of different HDAC subtypes that exert multiple functions in cellular mechanisms [27]. The landscape and heterogeneity of different HDAC subtypes in PCa and their role in tumorigenesis is still under investigation. Therefore, first-generation anti-HDAC agents are mainly pan-inhibitors targeting a variety of HDACs. However, the lack of specificity of pan-HDAC inhibitors makes it difficult to predict and assess the exact impact of the epigenetic drug on tumorigenesis, and results in frequent and severe side effects [136]. To avoid the high toxicity and heterogeneous effects of HDAC inhibitors, current research focuses on the development of selective inhibitors for specific HDAC subtypes [27]. In addition, the effect of HDAC inhibitors is particularly dose- and cell line-dependent, adding additional factors to consider when treating patients with HDAC inhibitors [21]. More extensive genomics-based selection of patients, for example, with high expression levels—or mutational status—of certain epigenetic enzymes, may enrich for patients who experience benefit from HDAC inhibitors treatment. 

Although HDAC inhibitors are the most studied epigenetic modifications in PCa, epigenetic alterations other than histone acetylation also have major impacts on prostate tumorigenesis. The increased expressions of epigenetic modifiers, such as KDM1A, BRD4 and EZH2, are common epigenetic hallmarks of PCa, and they are currently being investigated as epigenetic drug targets. The development of novel inhibitors targeting different epigenetic modifications also provides exciting opportunities for combination therapies. For example, anti-EZH2 agents exert promising new effects on prostate tumorigenesis when combined with AR inhibitors or immunotherapy [22,109].

Aberrant epigenetic modifications are promising novel drug targets for mCRPC, and further analysis of epigenetic mechanisms may help to elucidate which aberrant epigenetic modifications play a critical role in PCa, and this may also lead to the discovery of novel predictive biomarkers to aid in patient stratification. This is of particular relevance, as optimal patient selection in clinical trial design increases the likelihood of trial success, and at the same time, limits any potential adverse effects of the treatment on these patients, who may experience benefits from the drug. A more thorough analysis the PCa epigenome will help to better understand potential resistance mechanisms to epigenetic drugs and drive the development of improved epigenome-targeting agents. However, due to the diverse and relatively small effects of epigenetic agents as monotherapies, we believe that their use in combination with other therapies, especially together with AR inhibitors or immunotherapies, is a more realistic scenario for the near future. Finally, since immunotherapies have been particularly unsuccessful in PCa, we hypothesize that the use of epigenetic drugs could lead to major advances by reprogramming this immunologically “cold” tumor type.

Since prostate cancer and its microenvironment are intrinsically heterogeneous, intra-tumoral and inter-tumoral heterogeneity pose challenges [137]. Moreover, this heterogeneity is challenging to model in cell-culture models. Therefore, deeper analyses of the intra-tumoral heterogeneity of epigenetic factions, in relation to epigenetic therapy efficacy, would allow for a better understanding of clonal selection and treatment-induced epigenetic plasticity. A deeper understanding of the complex interplay between epigenetic modifications and the tumor microenvironment may provide greater mechanistic insights, which will enable the development of well-thought-out therapeutic strategies in the future and a new class of drugs for the treatment of mCRPC.

## Figures and Tables

**Figure 1 cancers-16-00294-f001:**
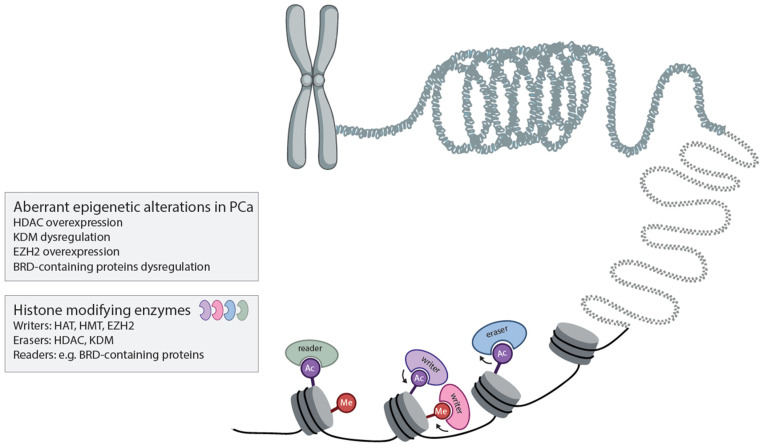
Graphic overview of important epigenetic modifications affecting PCa. Epigenetic modifications impact prostate tumorigenesis. Aberrant histone modifications affecting PCa are associated with histone acetylation and methylation. The modification of histone tails requires epigenetic writers (HAT, HMT, EZH2), depicted in pink and purple, depositing post-translational modifications at histone tails and epigenetic erasers (HDAC, H/KDM), depicted in blue, and removing them. Epigenetic readers (e.g., BRD-containing proteins), depicted in green, exert a variety of different catalytic and scaffolding functions. (HAT, histone acetyl transferase; HMT, histone methyl transferase; HDAC, histone deacetylase; H/KDM, histone lysine demethylase; BRD, bromodomain; EZH2, Enhancer of Zeste 2). Image adopted from BioRender.com.

**Figure 2 cancers-16-00294-f002:**
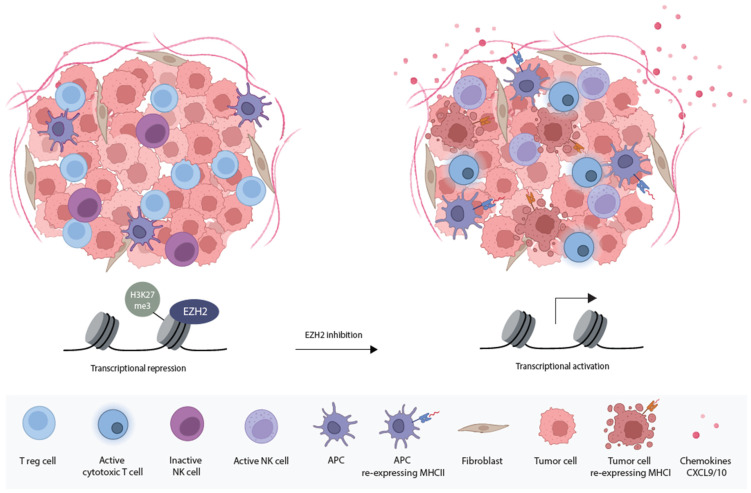
Graphic overview of the effect of EZH2 on the tumor and TME. The overexpression of EZH2 in cancer and cancer-associated cells promotes oncogenic immune evasion. EZH2 upregulation is involved in the dysregulation of MHC classes I and II, the maintenance of Treg cells, the suppression of pro-immunogenic CXCL9/10 chemokines and the inhibition of NK cell activation and their cytolytic activities. The inhibition of EZH2 could lead to a disturbance in Treg cell activity, a rise in CXCL9/10 chemokine expression and the re-expression of MHC classes I and II. These effects could increase cytotoxic T-cell activation and infiltration. EZH2 inhibitors could also enhance NK cell activation and restore their cytolytic activities. Therefore, EZH2 inhibition could decrease immunosuppression and sensitize tumors to immunotherapies. Adopted from BioRender.com.

**Table 1 cancers-16-00294-t001:** Summary of epigenome-targeting agents mentioned in the review (i = inhibitor).

Epigenetic Drug	Molecular Target	Cancer Type	Phase	References
Romidepsin	HDAC1/2i	Chemotherapy naïve mCRPC	Phase II	Molife et al. [68]
Vorinostat	HDACi	Progressive mCRPC after 1 prior chemotherapy treatment	Phase II	Bradeley et al. [69]

**Table 2 cancers-16-00294-t002:** Summary of epigenome-targeting agents mentioned in the review and their respective use in combination therapies (i = inhibitor, ICB = immune checkpoint blockade).

Epigenetic Drug	Molecular Target	Combination	Type	Cancer Type	Phase	References
Panibostat	Pan-HDACi	Bicalutamide	Anti-androgen	Recurrent PCa	Phase I/II	Ferrari et al. [70]
CPI-1205	EZH2i	Enzalutamide/Abiraterone/Prednisone	AR/androgen biosynthesis inhibitor/glucocorticoid receptor agonist	mCRPC patients	Phase Ib/II	Taplin et al. [71]
Tazometostat	EZH2i	Enzalutamide/Abiraterone/Prednisone	AR/androgen biosynthesis inhibitor/glucocorticoid receptor agonist	mCRPC patients	Phase I	Abida et al. [72]
ZEN-3694	BETi	Enzalutamide	ARi	CRPC patients	Phase II	Aggarwal et al. [73]
CCS1477	P300/CBPi	Enzalutamide/Abiraterone/Darolutamide/Olaparib/Atezolizumab	AR/androgen biosynthesis inhibitor/PARPi/IgG1 ab	Advanced solid tumors	Phase I/II	Bono et al. [74]
CPI-1205	EZH2i	Ipilimumab	Anti-CTLA4 ICB	Advanced solid tumors	Phase I/II	ClinicalTrial.gov [75]

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
