# Peer review of "It Takes Two to Tango: The Interplay between Prostate Cancer and Its Microenvironment from an Epigenetic Perspective"

_cancers, 2024, doi:10.3390/cancers16020294_

Round 1

Reviewer 1 Report

Comments and Suggestions for Authors

This is an excellent review and I commend the excellent work by the authors for bringing reviewing this important topic.

Thanks

Author Response

Response:

We thank this reviewer for time spent on reviewing our manuscript. We highly appreciate this reviewer’s judgement that the manuscript is excellent and that an important topic is reviewed.

Reviewer 2 Report

Comments and Suggestions for Authors

The manuscript “It takes two to tango: the interplay between prostate cancer and its microenvironment from an epigenetic perspective” is a review article regarding the current state-of-art of epigenetic alterations in prostate cancer and its TME promoting prostate tumorigenesis, and the impact of epigenome-targeted-therapies.

I appreciate the work performed by authors since the manuscript is generally well-written, provides a good view on the topic and can be of interest for the readers. However, there are important concerns that authors must address in order to consider the manuscript suitable for publication:

1.    The introduction section should include more information about prostate cancer, in order to provide a proper background to the readers.

2.    The tables within the manuscript are images that were wrongly stretched, resulting in a poor quality of the image with consequent difficulties of reading.

3.    Tables should include a column with the corresponding reference.

4.    There is an inconsistent formatting (font, size etc..) through the text. For example, in the end of the paragraph 1.2; please reconcile.

5.    The main concern is that this manuscript completely ignores a master regulator of cell epigenetics as well as tumor microenvironment, the enzyme nicotinamide N-methyltransferase (NNMT) which can influence a number of enzymes involved in epigenetics, such as histone deacetylases sirtuins (PMID: 36829935; PMID: 36817086; PMID: 31043742).

This enzyme has been reported to be overexpressed in a variety of solid tumors including prostate cancer, where it contributes to the tumorigenicity and aggressiveness, as recently reviewed. Since NNMT can affect NAD homeostasis, NAD-dependent enzymes and concentration of SAM, it has a great impact on epigenetics, as demonstrated by Ulanovskaya et al. (PMID: 23455543). Notably, a number of NNMT inhibitors are already available, and their use is a promising strategy for targeted therapy in cancer (PMID: 34572571; PMID: 34704059; PMID: 34424711; PMID: 36104373).  All these considerations should be included since they would let the manuscript cover totally the topic of possible epigenetic targets in prostate cancer.

Comments on the Quality of English Language

Minor editing of English language required.

Author Response

The manuscript “It takes two to tango: the interplay between prostate cancer and its microenvironment from an epigenetic perspective” is a review article regarding the current state-of-art of epigenetic alterations in prostate cancer and its TME promoting prostate tumorigenesis, and the impact of epigenome-targeted-therapies.

I appreciate the work performed by authors since the manuscript is generally well-written, provides a good view on the topic and can be of interest for the readers. However, there are important concerns that authors must address in order to consider the manuscript suitable for publication:

Response:

We thank reviewer #2 for concluding that the review article is well-written and the topic can be of interest to readers. Moreover, we thank this reviewer for the suggestions to improve the manuscript.

  1. The introduction section should include more information about prostate cancer, in order to provide a proper background to the readers.

Response:

We agree with the reviewer, that the clinical background on prostate cancer needed to be expanded. Therefore, we extended the Introduction section with general information on prostate cancer (page 4).

  1. The tables within the manuscript are images that were wrongly stretched, resulting in a poor quality of the image with consequent difficulties of reading.

Response:

We apologize for this issue. For the revised version, all Tables and Figures are now presented as separate pdf files, resolving this issue.

  1. Tables should include a column with the corresponding reference.

Response:

This was a clear omission, and we thank the reviewer for highlighting this. Both Table 1 and 2 now have a column listing the corresponding references.

  1. There is an inconsistent formatting (font, size etc..) through the text. For example, in the end of the paragraph 1.2; please reconcile.

Response:

All inconsistencies in formatting are removed, and we thank the reviewer for highlighting this inconsistency.

  1.  The main concern is that this manuscript completely ignores a master regulator of cell epigenetics as well as tumor microenvironment, the enzyme nicotinamide N-methyltransferase (NNMT) which can influence a number of enzymes involved in epigenetics, such as histone deacetylases sirtuins (PMID: 36829935; PMID: 36817086; PMID: 31043742). 

This enzyme has been reported to be overexpressed in a variety of solid tumors including prostate cancer, where it contributes to the tumorigenicity and aggressiveness, as recently reviewed. Since NNMT can affect NAD homeostasis, NAD-dependent enzymes and concentration of SAM, it has a great impact on epigenetics, as demonstrated by Ulanovskaya et al. (PMID: 23455543). Notably, a number of NNMT inhibitors are already available, and their use is a promising strategy for targeted therapy in cancer (PMID: 34572571; PMID: 34704059; PMID: 34424711; PMID: 36104373).  All these considerations should be included since they would let the manuscript cover totally the topic of possible epigenetic targets in prostate cancer.

Response:

We fully agree with this reviewer that NNMT cannot be ignored in the manuscript, and apologize for the omission. We therefore included an introduction on this subject on page 6-7, NNMT inhibitors on page 9 and the relation between NNMT and immunity on page 15. The suggested references have been added to the review.

  1. Minor editing of English language required.

Response:

Typos  and grammar issues have now been corrected throughout the manuscript.

Reviewer 3 Report

Comments and Suggestions for Authors

This review by Zaalberg A et al. gives an update of drug inhibition of DNA modifying enzymes such as HDAC and EZH2 in treating advanced prostate cancer. These factors alter the epigenome leading to gene expression perturbation in the cancer cells and cells in the tumor microenvironment. They may cause immune suppression through downregulation of MHC molecules in tumor antigen presentation, inhibit functional maturation of tumor killing cytotoxic T cells, NK cells. 

A number of clinical trials are listed testing drugs alone or more frequently in combination with AR targeting. None have proven effective to date with partial response, short-lived stable disease. The drugs are toxic causing a range of adverse side effects. Not unexpected since the targeted gene products like HDAC are undoubtedly key in normal cell types to modulate expression, for example, leading to functional differentiation. 

The authors could have addressed that one of the major challenges is to model the heterogeneity of the human disease comprising of multiple cancer cell types with different gene expression. In addition, the percentage of these biomarkers in tumors is largely unknown/unreported. For example, PTEN is found in 20% primary tumors, in 50% mCRPC. PTEN drugs would therefore be “effective” in half of the mCRPC patients. Would this lead to selection of non-PTEN tumors? Even in a single patient, there may be different cancer cell types as shown in xenografts (adenocarcinoma, non-adenocarcinoma, small cell carcinoma) established from excised tumor samples. Other challenges include lack of mechanistic detail in how these molecules affect biological processes such as immune response/suppression, how translatable are results obtained from cell lines and mouse models, which mainly study single cancer cell types at a time (suppose one could combine PTEN and non-PTEN tumors in one model, or EZH2/non-EZH2 tumors, or other combinations). The oft caveat as stated is that more research remains to be undertaken.

For an interested patient reading this it would be quite disappointing or even depressing. At present, these trials seem to lead to nowhere near a cure.

Author Response

Response:

We thank this reviewer for the comments. Indeed, thus far trials evaluating drugs modulating epigenetic factors failed to demonstrate a clinical response in un-preselected patient populations, and genomics-based patient selection for trial enrollment -as referred to by the reviewer- may be a powerful approach to potentially improve success rate of such clinical trials. We have updated the text accordingly (page 15 of the revised manuscript).

Intra-tumor heterogeneity is a known potential source of disease relapse, and we have  expanded the discussion section ( page 16) to better highlight this current largely-overlooked aspect in clinical trials.

Regarding cure-rates: sadly, as is the case for practically all cancer types, metastatic prostate cancer simply cannot be cured as this stage, by any drug. This simple fact cannot be brought differently. We do feel the discussion section highlights specific opportunities of novel drug combinations, with epigenetic therapeutics taking central stage, that may provide durable response in pre-selected patient populations, that we hope also gives a level of hope and outlook towards better treatment in the future, for the interested patient who would read our manuscript.

Reviewer 4 Report

Comments and Suggestions for Authors

The manuscript by A. Zaalberg et al. reviews current knowledge and approaches for prostate cancer treatment using epigenetic modifiers. The manuscript is generally well-written but has some repetitions and could be - improved by reconsidering the overall structure. By this I mean:

- subchapters 1.12-1.13 could be separated into chapter "2" as they describe general approaches to enhance tumor immunogenicity by epigenetic modifications, and not specifically processes for prostate cancer;

-subchapters 1.7. 1.10 and 1.11 could be integrated into others;

- tables 1 and 2 could be united to have an overall view. Table 2 header row should be reviewed to avoid duplicate names. Maybe "molecular target" or any more specific description name could be used instead of the left "Kind" column?

- This manuscript could reach a wider audience and become more readable if authors could critically review the use of abbreviations and reduce it as much as possible.

- Figure 1 needs a more detailed description in the legend as it should be self-explanatory without reading the whole manuscript. The color coding in the "Histone modulators" box is not clear.

Comments on the Quality of English Language

The English of the manuscript is clear. Some editing could be done to avoid spelling problems with abbreviations (e.g. CPB vs. CBP on page 5), and if possible, less use of abbreviations would improve the overall readability.

Author Response

The manuscript by A. Zaalberg et al. reviews current knowledge and approaches for prostate cancer treatment using epigenetic modifiers. The manuscript is generally well-written but has some repetitions and could be - improved by reconsidering the overall structure. By this I mean:

Response

We thank the reviewer for the suggestions to improve the manuscript.

- subchapters 1.12-1.13 could be separated into chapter "2" as they describe general approaches to enhance tumor immunogenicity by epigenetic modifications, and not specifically processes for prostate cancer;

-subchapters 1.7. 1.10 and 1.11 could be integrated into others;

Response

We thank the reviewer for this constructive remark. However, given the structure we designed of the manuscript, we feel the readibility and ‘flow’ of the story would be negatively affected if we change this. We hope the reviewer finds this acceptable.

- tables 1 and 2 could be united to have an overall view. Table 2 header row should be reviewed to avoid duplicate names. Maybe "molecular target" or any more specific description name could be used instead of the left "Kind" column?

Response

Thank you for your suggestion, we changed the header to “molecular target”. The reason we did not merge table 1 and 2 is to clarify that table 1 is about monotherapy, and table 2 describes combination therapies.

- This manuscript could reach a wider audience and become more readable if authors could critically review the use of abbreviations and reduce it as much as possible.

Response

We thank the reviewer for this suggestion. Following the reviewers’ advice, we tried to limit the number of abbreviations as much as possible.